# Hypernetworks for image recontextualization

**Maciej Zieba**
Wroclaw Tech., Tooploox, eBay Inc.
`maciej.zieba@tooploox.com`

**Jakub Balicki**
Tooploox, eBay Inc.

**Tomasz Drozdz**
Tooploox, eBay Inc.

**Konrad Karanowski**
Tooploox, eBay Inc.

**Pawel Lorek**
Wroclaw University, Tooploox, eBay Inc.

**Hong Lyu**
eBay Inc.

**Aleksander Skorupa**
Tooploox, eBay Inc.

**Tomasz Trzcinski**
Warsaw University of Technology,
IDEAS NCBR, Tooploox, eBay Inc.

**Oriol Caudevilla Torras**
eBay Inc.

**Jakub M. Tomczak**
Eindhoven Univ. of Tech., Tooploox, eBay Inc.

## Abstract

Image recontextualization, the task of placing a subject from an image into a new context to serve a specific purpose, has become increasingly important in fields like art, media, marketing, and e-commerce. Recent advancements in deep generative modeling, such as text-to-image and image-to-image synthesis via diffusion models, have significantly improved recontextualization capabilities. However, current methods, like DreamBooth and LoRA, require time-consuming fine-tuning per individual image, resulting in inefficiencies and often suboptimal outputs. Other approaches to recontextualization, like MagicClothing, require reorganization of the architecture of the base model and a time-consuming training process in a particular domain. In this work, we propose HyperLoRA, a novel framework that leverages hypernetworks to predict LoRA parameters, allowing for more efficient image recontextualization without the need for image-specific fine-tuning. HyperLoRA utilizes domain pairs of context images and target objects, enabling instant adaptation to new contexts while significantly reducing computational costs. Our method outperforms traditional techniques by offering more accurate adjustments, broader applicability across multiple modalities (e.g., text, video, sound, and structured data), and scalable deployment. Experimental results demonstrate the effectiveness of our approach in garment-to-model recontextualization, highlighting the potential for broader applications.

## 1   Introduction

*Image recontextualization* is a task that involves taking an image from its original context and presenting it in a new or different setting to serve a new purpose. This problem is typically encountered in art, media, and communication, and it is relevant to marketing and e-commerce. For instance, putting a photo of a purse captured by a seller in the context of a luxurious brand shopping window. In recent years, the concept of image recontextualization has gained more momentum due to deep generative modeling [1], particularly image-to-image and text-to-image synthesis models like diffusion models. An example of such an approach is DreamBooth [2] which fine-tunes a large pre-trained text-to-image diffusion model to enable personalization and subject recontextualization. This model

38th Conference on Neural Information Processing Systems (NeurIPS 2024).

takes a few images of a subject and then generates new images contextualizing the subject in different environments, views, or poses while preserving key visual features.

The problem in image recontextualization with pre-trained diffusion models is the necessity of fine-tuning. Fine-tuning of foundation models to specific tasks is a ubiquitous practice in machine learning, particularly in domains such as computer vision [3, 4, 5, 6] and natural language processing [7, 8]. Existing methods that tackle this problem, such as already mentioned DreamBooth [2], or LoRA [7], require fine-tuning per single image, which requires significant time and extensive prompting, while the results are often of low quality or do not match a given prompt.

Recent techniques, such as IP Adapter [9], enhance base architectures with additional, trainable cross-attention layers to preserve contextual details during recontextualization. However, these layers do not integrate context image information with textual features. Other methods, like MagicClothing [10], necessitate the design of specialized architectures to address the problem.

Here, we focus on improving fine-tuning of foundation models for image recontextualization. Unlike traditional methods that fine-tune models individually for each task or image, our approach leverages domain pairs comprising context images and target objects in desired contexts. For this purpose, we propose to utilize hypernetworks [11, 12] to predict LoRA [7] parameters, thus, we refer to our approach as *HyperLora*. The use of hypernetwork allows us to keep a single, shared neural network for outputting weights of a diffusion model. As a result, we eliminate the need for fine-tuning individual images, thereby significantly reducing computational costs and time requirements.

Our approach offers several advantages over existing methods. Firstly, it enables instant adaptation to new context images without the need for fine-tuning, resulting in more efficient and scalable model deployment. Secondly, we demonstrate how to effectively combine features from both text and context image modalities without requiring additional architectural designs or training in the base model. Additionally, our method is not limited to image data but can be applied across multiple modalities, including text, videos, sound, and structured data (e.g., point clouds [13]). Eventually, our hypernetwork provides a shared and unified representation for multiple contexts and data modalities.

The contributions of the paper are as follows: **(i)** we propose a new framework for predicting LoRA parameters in a parallel manner for new contexts, **(ii)** we outline how to formulate and train hypernetworks for image recontextualization, **(iii)** we show how to effectively combine textual and context image features without modifying the base model architecture, **(iv)** we present experimental results demonstrating the effectiveness of our approach on garment-to-model image recontextualization.

## 2 Related work

**Fine-tuning**   Low-rank adaptation LoRA [7] is one of the most common fine-tuning techniques, initially adopted to Large Language Models. The idea behind this approach is to utilize two trainable low-rank matrices, that create the high-rank matrix after multiplication. This matrix is further added to the frozen matrices used in transformer layers and trained in a gradient-based manner. Thanks to the special initialization of the low-ranked matrices, the initial values of the matrix components are equal to 0, which implies that the values do not have an impact on the model at the beginning of the training. VeRA [8] is one of the possible extensions of LoRA, where thanks to specific decomposition, it is sufficient to train vectors, instead of low-rank matrices. Various extensions of VeRA are extensively studied in the work presented in [14].

Textual inversion [3] is a basic technique for fine-tuning diffusion models like Stable Diffusion [15]. The main idea behind this approach is to train the text embedding that represents a specific concept represented by a given image in a gradient-based procedure. Extended textual inversion [4] enriches the idea with multiple textual embeddings injected into different attention layers of the U-Net architecture. NeTI [5] is another method based on textual inversion that optimizes the mapping network instead of directly training the textual embeddings. One of the most common techniques for model fine-tuning is DreamBooth [2]. This method uses a unique textual token to represent the concept of the image and fine-tunes the entire U-Net architecture. In addition, the model uses specific regularization to preserve the generalization features. In a follow-up, [9] proposed HyperDreamBooth to adapt the idea of DreamBooth to fast personalization. This approach utilizes hypernetworks but differs from our method by a large margin. First, we formulate a parallelized hypernetwork, while the HyperDreamBooth uses transformer-based architecture. Second, our approach does not need a

fine-tuning stage of LoRA parameters after training the hypernetwork. Finally, HyperDreamBooth is focused on personalization aspects, while our approach is more general and can utilize various modalities as context information.

**Recontextualization** IP Adapter [6] is a multitask approach that can also be applied for recontextualization applications. The authors introduce effective and lightweight adapter to achieve image prompt capability for the pre-trained text-to-image diffusion models. The key design of our IP-Adapter is decoupled cross-attention mechanism that separates cross-attention layers for text features and image features. Another approach, MagicFusion [16], incorporates saliency-aware noise blending to preserve the details from contextual image during the fine-tuning approach. To preserve the details of the item, MagicClothing [10] uses two parallel UNets, one for processing garment features and the other for model generation. Compared to the described reference methods, our approach does not require problem-specific architectures and effectively combines textual and contextual information within a single cross-attention layer with no further fine-tuning.

# 3 Background

## 3.1 Diffusion Models

**Diffusion models** Diffusion models [17, 18, 15] have risen to prominence within the generative model area, setting new baselines in the field of image generation [19, 20, 21]. Their utility extends to various practical applications such as enhancing image quality [22, 23], compressing images [24], image manipulation [25, 26], colorization [27, 28], style transfer [29, 30, 28], among others.

Diffusion models are a class of generative models designed to closely approximate the original data (images, or images conditioned on some additional information) distribution $q(\mathbf{x}_0)$ from image space $\mathcal{I}$ with model's distribution $p_\theta(\mathbf{x}_0)$:

$$p_\theta(\mathbf{x}_0) = \int \left[ p_\theta(\mathbf{x}_T) \prod_{t=1}^{T} p_\theta^t(\mathbf{x}_{t-1}|\mathbf{x}_t) \right] d\mathbf{x}_{1:T}, \tag{1}$$

where $\mathbf{x}_{1:T} = \mathbf{x}_1, \ldots, \mathbf{x}_T$ are variables of a forward Markov chain such that $\mathbf{x}_t = \sqrt{\alpha_t}\mathbf{x}_0 + \sqrt{1-\alpha_t}\boldsymbol{\varepsilon}$, where $\alpha_t$ is a so-called noise scheduler (a hyperparameter) such that $0 = \alpha_T < \alpha_{T-1} < \cdots < \alpha_1 < \alpha_0 = 1$ assuring that $\mathbf{x}_t$ converges to a Gaussian noise $\boldsymbol{\varepsilon} \sim \mathcal{N}(\mathbf{0}, \boldsymbol{I})$ as $t$ approaches $T$.

**Latent Diffusion Models** Latent Diffusion Models (LDMs) [15, 31] improve the efficiency (and the quality) of diffusion models by operating in a latent space. LDMs use some (variational) autoencoder pre-trained on a large collection of images, with an encoder $\mathcal{E}$ that maps an image $\mathbf{x}$ into latents, $\mathbf{z} = \mathcal{E}(\mathbf{x})$, and a decoder $\mathcal{D}$ such that $\mathcal{D}(\mathcal{E}(\mathbf{x})) \approx \mathbf{x}$. The diffusion model is then trained to produce representations of images in a latent space and could be seen as a marginal distribution over latents. For an image $\mathbf{x}_0$, the diffusion process adds noise to latent image $\mathbf{z}_0 = \mathcal{E}(\mathbf{x}_0)$ producing a noisy latent image $\mathbf{z}_t$. The amount of noise increases with $t \in \{1, \ldots, T\}$. The input image $\mathbf{x} = \mathbf{x}_0$ can be conditioned on some additional information $\mathbf{y}$, e.g., text prompt (text instruction), other image(s), etc. from space $\mathcal{C}$. By $D_{\text{train}}$ we denote a training set o pairs $(\mathbf{x}, \mathbf{y}) \in \mathcal{I} \times \mathcal{C}$. Let $\tau_\theta(\mathbf{y})$ be a model that maps conditioning input information $\mathbf{y}$ from space $\mathcal{C}$ into a vector of predefined dimension. The model's loss is defined as:

$$L_{\text{LDM}} = \mathbb{E}_{(\mathbf{x}_0, \mathbf{y}) \sim D_{\text{train}}, \mathbf{z}_0 = \mathcal{E}(\mathbf{x}_0), t \sim \text{Unif}(\{1,2,\ldots,T\}), \boldsymbol{\varepsilon} \sim \mathcal{N}(\mathbf{0}, \boldsymbol{I})} ||\boldsymbol{\varepsilon} - \varepsilon_\theta(\mathbf{z}_t, t, \tau_\theta(\mathbf{y}))||_2^2, \tag{2}$$

where $\boldsymbol{\varepsilon}$ is a Guassian noise, $t$ is a time step, $\mathbf{z}_t$ is the latent noised up to time $t$, i.e., $\mathbf{z}_t = \sqrt{\alpha_t} + \sqrt{1-\alpha_t}\boldsymbol{\varepsilon}$ and $\varepsilon_\theta$ is a denoising network. Roughly speaking, the objective is to *correctly* remove the noise which was added. Networks $\tau_\theta$ and $\varepsilon_\theta$ are jointly optimized to minimize $L_{\text{LDM}}$. At inference time, a random noise $\mathbf{z}_T \sim \mathcal{N}(\mathbf{0}, \boldsymbol{I})$ is sampled and iteratively denoised to produce latent image $\mathbf{z}_0$, which is afterward transformed into an image via decoder, i.e., $\mathbf{x} = \mathcal{D}(\mathbf{z}_0)$.

## 3.2 Fine-tuning of LDMs for recontextualization

**Problem formulation** In the case of a task for which only a few samples are available, e.g., generating images of a given *style* represented by a set of a few images, we could train a diffusion

model from scratch. However, especially if the number of images is limited, *fine-tuning* a large diffusion models outperforms training the model from scratch. Roughly speaking, the procedure starts with a pre-trained model with parameters $\theta$ and gradually changes their values to $\theta'$. The rationale is to adjust the model to the task at hand to improve its performance. This approach leverages the general capabilities learned during the initial training phase and adjusts the model's parameters mildly to better capture the nuances of the new data, thus, enhancing its ability to generate or process data.

Fine-tuning as explained above is widely used to tailor generic models to specialized applications without the need for training a model from scratch. In this paper, we focus on a specific example, namely, we fine-tune a diffusion model for recontextualization using a collection of garments. Our **goal** is then to generate a realistic image of a human model wearing a specific garment keeping all the details (fabric, color, logos, etc.) of the garment. Formally, let us denote a number of new pairs $(\mathbf{x}, \mathbf{y})$ as $D_{\text{train}}^{\text{FT}}$, then the fine-tuning loss can be defined as follows:

$$L_{\text{FT}} = \mathbb{E}_{(\mathbf{x}_0', \mathbf{y}') \sim D_{\text{train}}^{\text{FT}}, \mathbf{z}_0' = \mathcal{E}(\mathbf{x}_0'), t \sim \text{Unif}(\{1, 2, \dots, \text{T}\}), \boldsymbol{\varepsilon} \sim \mathcal{N}(\mathbf{0}, \boldsymbol{I})} ||\boldsymbol{\varepsilon} - \varepsilon_\theta(\mathbf{z}_t', t, \tau_\theta(\mathbf{y}'))||_2^2. \tag{3}$$

Losses $L_{\text{LDM}}$ and $L_{\text{FT}}$ are almost identical, the difference lies in the used empirical distribution and the initial weight values of the model. We assume that when optimizing the objective (3), we start with a pre-trained model, namely, $\theta$ resulting from (2), however, not all weights are optimized. For instance, DreamBooth [2] fine-tunes all parameters of the U-Net model $\varepsilon_\theta$, whereas Textual Inversion [3] (where the space $\mathcal{C}$ of conditional additional information is a set of prompts) fine-tunes a special token (not present in $\mathcal{C}$) – its embedding to be more precise – in the CLIP text encoder.

**Attention-based methods** An interesting line of research aims at modifying the text-to-image diffusion process through adjustments in cross-attention layers, allowing for more precise control over the images produced. In [32], the capabilities of this approach are demonstrated in an example of editing actual images in various scenarios. More specifically, the cross-attention block adjusts the network's latent features based on conditional features, such as text features in text-to-image diffusion models. For given text features $\mathbf{c} \in \mathbb{R}^{s \times d}$ and latent image features $\mathbf{f} \in \mathbb{R}^{h \times w \times l}$, a single cross-attention operation consists of $\mathbf{Q} = \mathbf{W}^q \mathbf{f}, \mathbf{K} = \mathbf{W}^k \mathbf{c}, \mathbf{V} = \mathbf{W}^v \mathbf{c}$. The operation then computes a weighted sum over the value features:

$$\text{Attention}(\mathbf{Q}, \mathbf{K}, \mathbf{V}) = \text{Sofmtax}\left(\frac{\mathbf{Q}\mathbf{K}^T}{\sqrt{d'}}\right)\mathbf{V}, \tag{4}$$

where projection matrices $\mathbf{W}^q$, $\mathbf{W}^k$ and $\mathbf{W}^v$ map the inputs to a *query*, *key* and *value feature* respectively. Here $d'$ is the output dimension of key and query features. The output of the attention layer is further transformed using the output linear projection layer parameterized by $\mathbf{W}^o$:

$$\mathbf{Z} = \mathbf{W}^o \text{Attention}(\mathbf{Q}, \mathbf{K}, \mathbf{V}). \tag{5}$$

The aim of fine-tuning is to update mappings from a given prompt to image distribution, and text-based features appear only in $\mathbf{W}^k$ and $\mathbf{W}^v$, that is why in [32] it is proposed to update only parameters of these two matrices.

**LoRA**. Many recent approaches, like LoRA [7], show the benefits of modifying all four $\mathbf{W}^q, \mathbf{W}^k, \mathbf{W}^v, \mathbf{W}^o$ matrices instead of fine-tuning the parameters of $\mathbf{W} \in \mathbb{R}^{d \times s}$ by updating their lower-dimensional representations (a product of two *low rank* matrices). LoRA keeps the pre-trained weights frozen and injects trainable decomposition matrices (into each layer). Formally, let $i = 1, \dots, L$ be the layer number. At layer $i$ we have four pre-trained matrices $\mathbf{W}_i^j, j \in \{q, k, v, o\}$. Let $r \ll \min(d, s)$. The update is constrained by representing the latter with a low-rank decomposition, namely:

$$\mathbf{W}_i^j + \Delta\mathbf{W}_i^j = \mathbf{W}_i^j + \mathbf{B}_i^j \mathbf{A}_i^j, \qquad i = 1, \dots, N, \quad j \in \{q, k, v, o\}, \tag{6}$$

where $\mathbf{B}_i^j \in \mathbb{R}^{d \times r}, \mathbf{A}_i^j \in \mathbb{R}^{r \times s}$. During training weights $\mathbf{W}_i^j$ are frozen, only $\mathbf{A}_i^j$ and $\mathbf{B}_i^j$ are trainable. For $\mathbf{h}_i^j = \mathbf{W}_i^j \mathbf{x}$, the modified forward pass is the following:

$$\mathbf{h}_i^j = \mathbf{W}_i^j \mathbf{x} + \Delta\mathbf{W}_i^j \mathbf{x} = \mathbf{W}_i^j \mathbf{x} + \mathbf{B}_i^j \mathbf{A}_i^j \mathbf{x}.$$

Moreover, $\mathbf{B}_i^j$ are initialized with zeros, i.e., $\mathbf{A}_i^j$ are initialized with random Gaussian noise, so that $\Delta\mathbf{W}_i^j = \mathbf{0}$ initially. In other words, the model kicks off from the original pre-trained matrices.

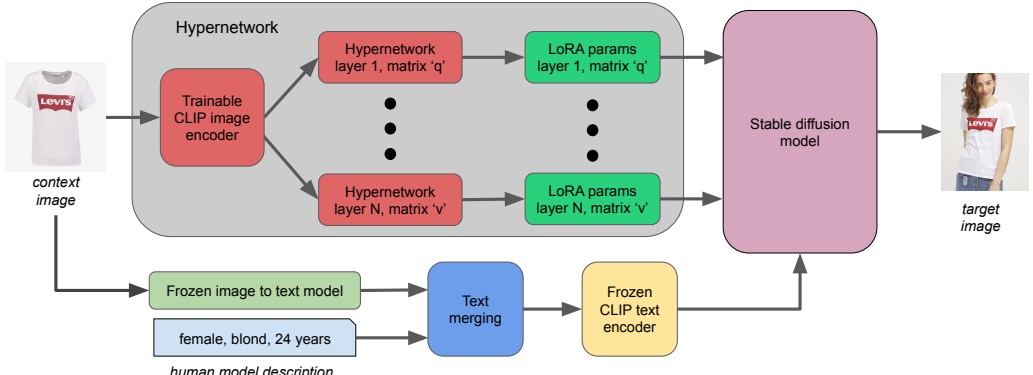

Figure 1: The overview of the image generation process using HyperLoRA. The context image is transformed using a trainable CLIP encoder. The output is further transformed by small-sized, parallel, shallow hypernetworks that predict the values of matrices $\mathbf{A}_i^j$ and $\mathbf{B}_i^j$ (LoRA parameters) for $i$-th layer and $j$-th matrix for cross-attention layers of a foundation model like Stable Diffusion. The textual description of the context image is generated using a pre-trained text-to-image model. Next, the semantic description of the target context (attributes of the model) and the context image are merged and tokenized with a frozen clip encoder. The predicted LoRA parameters are further used together with the embedded prompt containing the tokenized description of the image in the new context to generate the target image using Stable Diffusion.

## 4 Our Method

### 4.1 HyperModels

This work introduces HyperLoRA, a novel approach for recontextualization. Compared to reference approaches, like MagicClothing [10], our method works like a plug-in model and does not require modifying the parameters of the base diffusion model. Our solution also does not require additional cross-attention mechanisms like the IP Adapter [6], which injects the features of context image separately from the textual information. Instead, we utilize LoRA parameters to combine the textual and image context features together in the attention layers of the base diffusion model.

In our approach, we postulate the use of hypernetworks, the networks that predict the parameters of the other model to adapt it to a particular context. The hypernetwork takes a context image as input and predicts the parameters that are usually fine-tuned using standard procedures like LoRA. Thanks to this approach, the set of parameters is adjusted to the new context image without needing a fine-tuning procedure, reducing time significantly.

The rest of this subsection is organized as follows. First, we provide the architectural details of our approach. Second, we focus on how the model is trained using domain data. Finally, we describe how the trained model is utilized for recontextualization.

### 4.2 Architecture of HyperLoRA

The architecture of the proposed HyperLoRA model is provided in Figure 1. The contextual image $\mathbf{x}^c$ is fed to a hypernetwork $\mathcal{H}_\phi(\cdot)$ parameterized by $\phi$. $\mathcal{H}_\phi(\cdot)$ is composed of a trainable CLIP image encoder and shallow parallelized hypernetwork layers that predict individual LoRA low-ranked matrices $\mathbf{A}_i^j, \mathbf{B}_i^j$ for each of cross-attention matrices $j \in \{q, k, v, o\}$ on each layer $i \in \{1, \ldots, N\}$ of the denoising network. During training, all $\mathbf{W}_i^j$ are frozen. The predicted matrices $\mathbf{A}_i^j, \mathbf{B}_i^j$ are added to frozen $\mathbf{W}_i^j$ to the attention layers in the same manner as in LoRA-based fine-tuning, given by Eq. (6). The prompt for the target image is generated by merging the textual description of the context image with the description of the target context, e.g., attributes of a human model wearing a desired item given by the context image. For both image-to-text and merging operations, we use a pre-trained large language model, e.g., ChatGPT-4. The target image is generated using the input prompt in the same manner as the fine-tuned LoRA model.

### 4.3 Training HyperLoRA

To train HyperLoRA, we use a paired set composed of context images describing a concept and its corresponding target image, together with conditioning textual description (prompt). The training set $D_{\text{train}}$ consists of triples $(\mathbf{x}^{\text{c}}, \mathbf{x}^{\text{tg}}, \mathbf{y}^{\text{tg}})$, where $\mathbf{x}^{\text{c}}$ is the context image, $\mathbf{x}^{\text{tg}}$ is the corresponding target image and $\mathbf{y}^{\text{tg}}$ is the target prompt.

We apply training with respect to the parameters $\phi$ of $\mathcal{H}_\phi(\cdot)$ by minimizing the following loss function:

$$L_{\text{HyperLoRA}} = \mathbb{E}_{(\mathbf{x}^{\text{c}}, \mathbf{x}_0^{\text{tg}}, \mathbf{y}^{\text{tg}}) \sim \mathcal{D}, \mathbf{z}_0^{\text{tg}} = \mathcal{E}(\mathbf{x}_0^{\text{tg}}), t \sim \text{Unif}(\{1,2,...,\text{T}\}), \boldsymbol{\varepsilon} \sim \mathcal{N}(\mathbf{0}, \boldsymbol{I})} ||\boldsymbol{\varepsilon} - \varepsilon_\theta||_2^2, \tag{7}$$

where $\varepsilon_\theta \equiv \varepsilon_\theta(\mathbf{z}_t^{\text{tg}}, t, \tau_\theta(\mathbf{y}^{\text{tg}}), \mathcal{H}_\phi(\mathbf{x}^{\text{c}}))$, and $\mathcal{H}_\phi(\mathbf{x}^{\text{c}})$ provides the LoRA parameters to the denoising model $\varepsilon_\theta$.

Moreover, to emulate the zero-value condition for initialization of $\mathbf{B}_i^j$ matrices ($\mathbf{B}_i^j = \mathbf{0}$), we incorporate additional, trainable scaling parameter $\gamma$ (the same for all matrices, initially close to 0) and scale the matrices' values, i.e., in Eq. (6) the update is as follows:

$$\Delta \mathbf{W}_i^j = \gamma \mathbf{B}_i^j \mathbf{A}_i^j. \tag{8}$$

Thus, the trainable parameters of the model are: CLIP image encoder, shallow hypernetworks (each predicting values of matrices $\mathbf{A}_i^j$ and $\mathbf{B}_i^j$) and $\gamma \in \mathbb{R}$. The initial values of $\gamma$ are set close to 0. With such initialization, the values of $\mathbf{B}_i^j$ do not affect the model performance at the early stage of the training.

### 4.4 Recontextualization using HyperLoRA

Assuming a trained hypernetwork $\mathcal{H}_\phi(\cdot)$ and the context image $\mathbf{x}^{\text{c}}$ together with semantic description of target context, $\mathbf{y}^{\text{tgc}}$, we aim at generating the target image $\mathbf{x}^{\text{tg}}$. The hypernetwork takes the context image $\mathbf{x}^{\text{c}}$, $\mathcal{H}_\phi(\mathbf{x}^{\text{c}})$, and predicts the LoRA parameters as it was described in Sect. 4.2. The semantic description extracted from context image $\mathbf{x}^{\text{c}}$ and the description of target context, $\mathbf{y}^{\text{tgc}}$ are merged together to create the prompt for the target image $\mathbf{y}^{\text{tg}}$. Finally, the target image is generated using the prompt $\mathbf{y}^{\text{tr}}$ with the stable diffusion model with the LoRA weights predicted by hypernetwork $\mathcal{H}_\phi(\cdot)$.

## 5 Experiments

We evaluate our approach empirically and compare the results with selected reference methods adapted to contextualization tasks. First, we make the quantitative evaluation using metrics that are focused on maintaining consistency of generated and ground true images, comparing the results with reference approaches. Second, we focus on qualitative evaluation using both model variants with and without pose guidance.

**Dataset.** We use the VITON HD [33] dataset that contains image pairs representing a top clothing image and a frontal-view human model wearing the corresponding item. The original dataset comprises $11,647$ training and $2,032$ testing pairs.

**Evaluation Metrics.** Following the methodology from [2], we use the following evaluation metrics, which are mainly focused on preserving consistency between ground truth and generated photos, namely:

- *Context to generated.* (**C2G**). This metric represents the cosine similarity between the CLIP [34] embeddings of the context and the generated target images.

- *Target to generated.* (**T2G**). This metric is calculated similarly as **C2G**, but a ground-truth target image is used instead of a context one.

- *Target to generated difference* (**T2Gdiff**). This metric represents the absolute value of the difference between **C2G** and **C2T**, where **C2T** is the CLIP cosine similarity between ground-true context and target images.

Table 1: Quantitative comparison of HyperLoRa, Text HyperLoRA, and reference methods on VITON HD test set (best in **bold**, and ↑, ↓ represent whether we aim for maximizing or minimizing, respectively).

| | Without Poses | | | With Poses | | |
|---|---|---|---|---|---|---|
| | T2G ↑ | C2G ↑ | T2Gdiff ↓ | T2G ↑ | C2G ↑ | T2Gdiff ↓ |
| **HyperLoRA** | **0.8893** | 0.7738 | **0.0535** | **0.8899** | 0.7758 | 0.0497 |
| **HyperLoRA + IP Adapter** | 0.8495 | **0.8510** | 0.0658 | 0.8497 | **0.7991** | **0.0478** |
| **IP-Adapter** | 0.8060 | 0.8157 | 0.0693 | 0.8150 | 0.7968 | 0.0579 |
| **Magic Clothing** | 0.7504 | 0.6414 | 0.1278 | 0.8048 | 0.6745 | 0.1382 |

**Methods.** We compare our approach against selected reference methods, such as IP Adapter and MagicClothing. We selected those baselines because they utilize the entire training set to train the recontextualization method for a particular domain unlike standard fine-tuning methods (e.g., LoRA or Dreambooth) that utilize few images to fine-tune for a single item.

We adopt a two-stage training approach for our method. Initially, the model is trained using a static prompt, *'A woman wearing S\*'*. In the subsequent stage, the target prompt is generated by merging the descriptions of the conditioning image and the target context (attributes of the model wearing the item, see Figure 1). For tasks involving image-to-text conversion and text merging, we leverage ChatGPT-4. Two evaluation variants are considered for each model used in the experiments. The first, referred to as the standard approach, generates the target image based solely on the context image. The second variant incorporates pose guidance, generating a target image of the model in a specified pose while wearing the item from the context image. This is achieved by utilizing model weights integrated into the ControlNet [35] architecture. Additionally, we evaluate a variant of our model combined with IP Adapter to improve consistency between the generated and target images

## 5.1 Quantitative comparison

We employed the VITON HD test set [33], which consists of 2,032 images, for quantitative evaluation. For these images, we calculated the **C2G**, **T2G** (where higher values indicate better performance), and **T2Gdiff** (where lower values indicate better performance) metrics.

Table 1 presents a comparison of the computed metric values across all generated images. The results show that the **HyperLoRA** variants consistently outperform the baseline methods across all metrics, demonstrating superior performance. When utilizing poses aligned with the target images, we observe a slight reduction in the **T2Gdiff** score and a marginal improvement in the **T2G** metric. Additionally, the integration of the IP Adapter with the HyperLoRA model leads to an increase in consistency between the context and generated images, as reflected by the improved **C2G** metric.

## 5.2 Qualitative comparison

In Tables 2 and 3, we provide qualitative results comparing the performance of different methods, both without and with pose guidance. The methods based on HyperLoRA demonstrate superior preservation of contextual information compared to other techniques, particularly in simpler cases, and show mostly improved performance in moderately challenging cases. Additionally, the poses generated by our proposed methods exhibit a more natural appearance. Notably, the combination of **HyperLoRA** with the **IP Adapter** appears to offer enhanced preservation of item details. However, it is important to acknowledge that all methods evaluated exhibit some degree of inconsistency in preserving the details of certain logos across the experiments. This phenomenon is observed due to the fact that HyperLoRA compresses the input image into a small embedding vector, which may lead to losing important high-frequency details. Therefore, in future works, we will investigate incorporating the parallel Unet concept provided in MagicClothing into our approach, in order to preserve both high-level and low-level features.

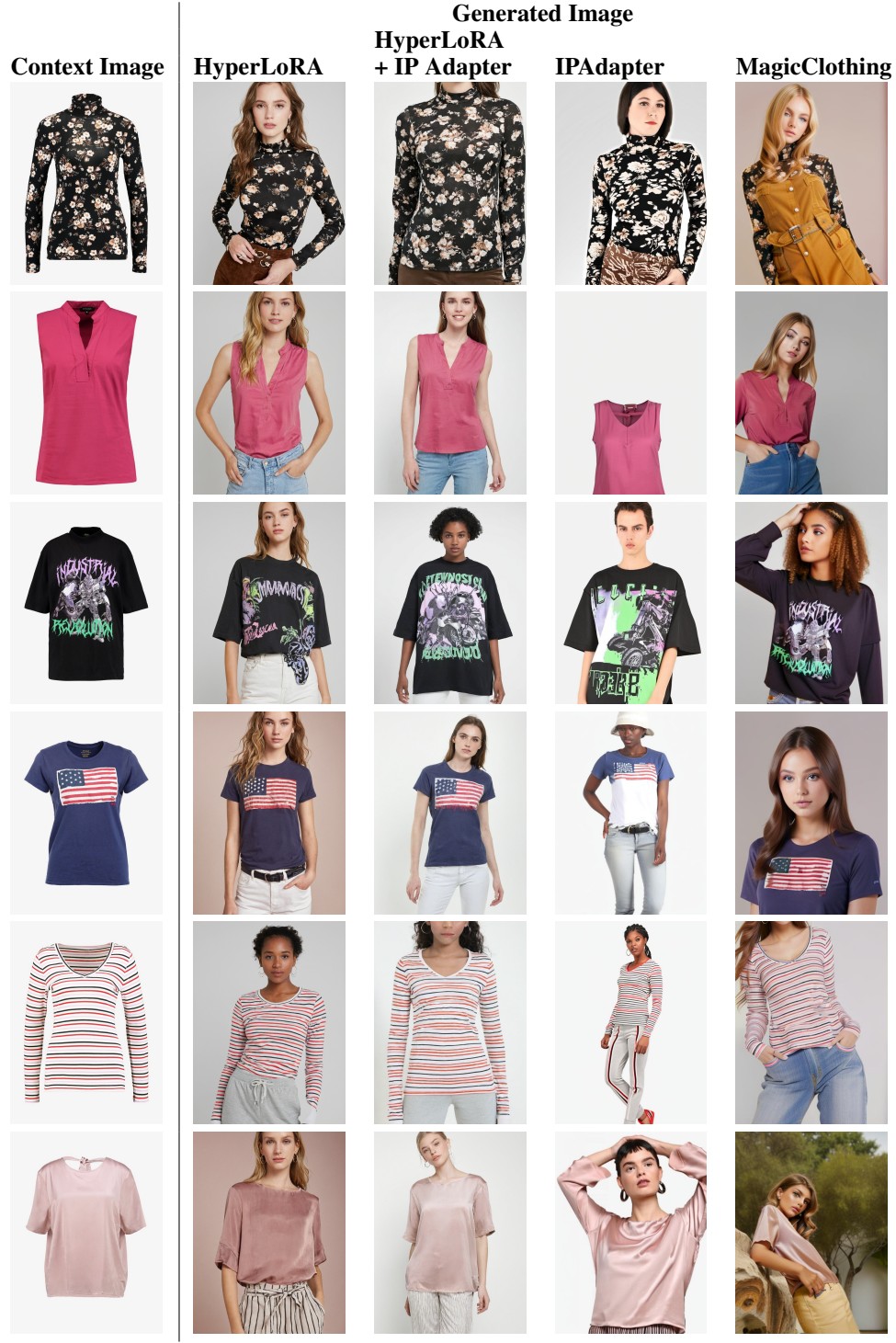

Table 2: The visual comparison of images generated by HyperLoRA, HyperLoRA + IP Adapter, and reference methods, conditioned on garments from the VITON test dataset.

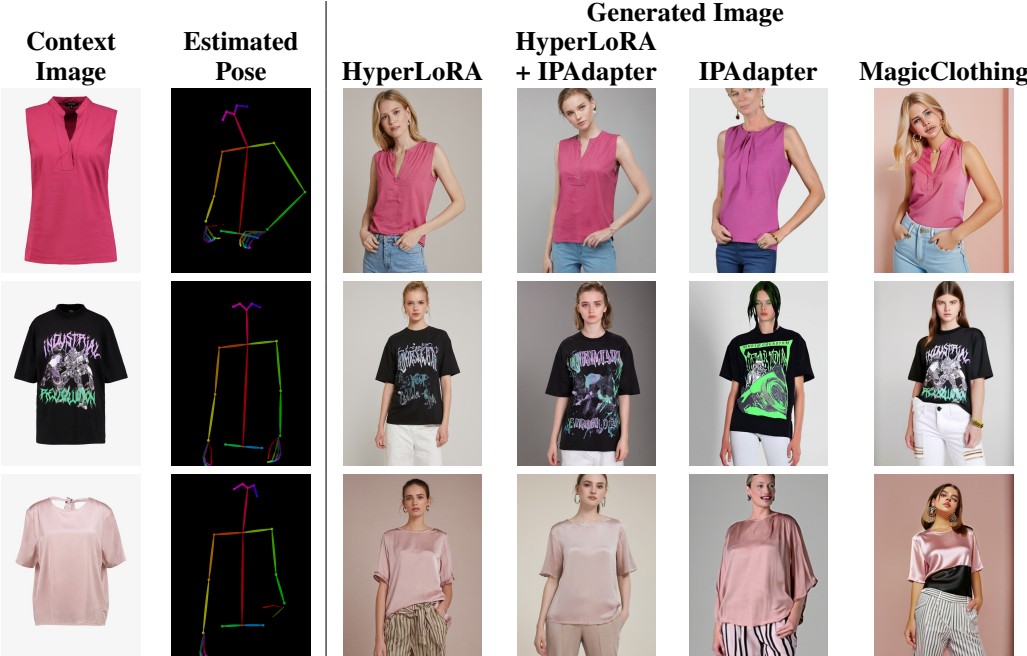

| Context Image | Estimated Pose | Generated Image | | | |
|---|---|---|---|---|---|
| | | HyperLoRA | HyperLoRA + IPAdapter | IPAdapter | MagicClothing |

Table 3: The visual comparison of images generated by the models with additional pose guidance.

# 6 Conclusions

The research introduces a novel method for instant recontextualization in fine-tuning foundation models, addressing shortcomings of existing approaches such as DreamBooth and LoRA. Unlike traditional methods, which require fine-tuning per single image and extensive prompting, the proposed approach leverages domain pairs and hypernetworks to predict parameters, eliminating the need for fine-tuning individual images and significantly reducing computational costs and time requirements. The method offers several advantages, including instant adaptation to new context images without fine-tuning, more accurate adjustments, and applicability across multiple modalities. Compared to methods that utilize context images like IP Adapter or MagicClothing our approach combines textual and image features in one attention mechanism and does not require any modifications and training of the base generative model. Experimental results demonstrate its effectiveness, particularly in garment-to-model recontextualization, highlighting its contributions to the field.

For future work, we would like to extend our method using a variety of modalities, including text, metadata, video, and speech. The adaptation of our model can be achieved simply by replacing the trainable image CLIP encoder with another type of encoding network suitable for such data. We also plan to extend the applications of HyperLoRA to other categories, considering the target and contested images in more sophisticated concepts.

The other dimension, where our method can be extended, touches the representation of the predicted parameters for our model. In our studies, we focus on predicting the values of matrices of low-rank matrices used in LoRA-based decomposition. However, our framework can be successively applied to other decomposition methods, like VeRA [8] or other approaches described in [14].

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
