# OpenReview forum: "Hypernetworks for image recontextualization"
_NeurIPS.cc/2024/Workshop/UniReps — UniReps_

### Official Review · Reviewer_aFCd · 2024-09-29
**HyperLoRA = Hypernetworks for LoRA**

**Rating:** 6
**Confidence:** 3

**Review:**

This paper proposes to predict LoRA parameters using a hypernetwork. This allows to improve contextualization in the generated images. Second, HyperLoRA offers a more efficient and scalable deployment because it enables instant adaptation to a particular context. Experiments show advantages of incorporating image and textual contexts to image generation using the stable diffusion model. Overall, HyperLoRA might be a practical method for recontextualization even though it has an incremental academia contribution.

---

### Official Review · Reviewer_eZcd · 2024-10-01
**Accept - Hypernetworks for image recontextualization**

**Rating:** 7
**Confidence:** 4

**Review:**

The paper proposes HyperLoRA, a novel approach leveraging hypernetworks for image recontextualization without the need for image-specific fine-tuning. One of its key strengths lies in its ability to reduce computational costs and time compared to existing methods like DreamBooth and LoRA, which require extensive fine-tuning for each image. Another strength is the model's adaptability across multiple modalities (e.g., text, video, sound), making it versatile for various recontextualization tasks. The experimental results demonstrate its superior performance in garment-to-model recontextualization, and the qualitative comparisons further showcase its effectiveness in preserving item details and generating natural-looking results, particularly when combined with pose guidance.

However, potential weaknesses include the limited exploration of applications beyond garment-to-model recontextualization. Additionally, while the paper highlights performance improvements over IP Adapter and MagicClothing, more detailed comparisons with additional state-of-the-art methods could provide a more comprehensive evaluation. Expanding experiments to include more complex, real-world scenarios and larger datasets would strengthen the argument for broader applicability. Overall, HyperLoRA shows great promise but could benefit from further validation in diverse contexts.

---

### Official Review · Reviewer_Gn7i · 2024-10-01
**Promising and Novel Approach for Image Recontextualization with Room for Further Comparison**

**Rating:** 6
**Confidence:** 3

**Review:**

## Summary

The paper proposes a novel method for image recontextualization, specifically focusing on the task of placing an object or subject from an image into a new, specified context. The method introduces a **hypernetwork** that generates parameters for **LoRAs** (Low-Rank Adaptations), which are then incorporated into a generative model (Stable Diffusion). Conditioned on a given input image, the hypernetwork predicts the necessary LoRA parameters, which, along with a textual description of the new context, guide the model to produce an image of the input subject within the new setting. The paper presents promising results on a garment-to-model recontextualization task and suggests potential applications in various domains beyond the demonstrated use case.

## Strengths and Weaknesses

Strengths:

- The paper is well-written and easy to follow.
- The method is novel, it is simple and it works. It is easy to imagine this method being applied to other modalities.

Weaknesses:

- In the **related work** section, there is limited discussion of methods that enable real-time conditioning based on a single image, such as **MagicClothing** and **IP-Adapter**. These approaches are relevant because they avoid the need for image-specific fine-tuning, similar to the proposed method. A more thorough comparison of these real-time techniques would strengthen the paper, especially in terms of how HyperLoRA differentiates itself from these methods.
- In Table 2 and 3, it seems that HyperLoRA makes some visible mistakes when recontextualizing the black t-shirt. On the other hand, MagicClothing seems to get it right. This may be due to the way HyperLoRA’s image encoder compresses the input image into a small embedding vector, potentially losing important high-frequency details. By contrast, MagicClothing’s U-Net architecture likely handles such details more effectively. A more detailed analysis of these errors and a comparison between the methods would add value to the paper.

## Questions/Suggestions

- The sentence in line 85 seems to contain an error: “[…] is a multitask approach **that solves** that also”. Consider revising this sentence to remove the redundancy.

Soundness: 3

Presentation: 3

Contribution: 3

---

### Decision · Program_Chairs · 2024-10-10

**Decision:**

Accept

**Comment:**

In light of the positive reviewers' feedback and relevancy of the submission, we are pleased to accept this paper for presentation at UniReps 2024. We kindly ask the authors to incorporate the reviewers' suggestions and feedback in the final camera-ready version of the manuscript.